# Subjective Wellbeing and Related Factors of Older Adults Nine and a Half Years after the Great East Japan Earthquake: A Cross-Sectional Study in the Coastal Area of Soma City

**DOI:** 10.3390/ijerph19052639

**Published:** 2022-02-24

**Authors:** Yuri Kinoshita, Chihiro Nakayama, Naomi Ito, Nobuaki Moriyama, Hajime Iwasa, Seiji Yasumura

**Affiliations:** 1Department of Living and Culture, Tohoku Seikatsu Bunka Junior College, 1-18-2, Nijino-oka, Izumi-ku, Sendai, Miyagi 981-8585, Japan; 2Department of Public Health, Fukushima Medical University School of Medicine, 1 Hikariga-oka, Fukushima 960-1295, Japan; nakac@fmu.ac.jp (C.N.); moriyama@fmu.ac.jp (N.M.); hajimei@fmu.ac.jp (H.I.); yasumura@fmu.ac.jp (S.Y.); 3Department of Radiation Health Management, Fukushima Medical University School of Medicine, 1 Hikariga-oka, Fukushima 960-1295, Japan; itonaomi@fmu.ac.jp

**Keywords:** disaster, tsunami, older adults, PGC morale scale, subjective wellbeing, Fukushima

## Abstract

This study examined older adults’ subjective wellbeing and related factors in the coastal area of Soma City nine and a half years after the Great East Japan Earthquake (GEJE). Data were collected from 65- to 84-year-old residents and 1297 participants via a questionnaire from October to November 2020. The participants were divided into two groups: housing complexes and non-housing complexes. The dependent variable was subjective wellbeing assessed via Lawton’s Philadelphia Geriatric Center Morale Scale (PGCMS). Using multivariate regression analysis, the factors most strongly related to a low PGCMS score for both groups were poor health conditions, difficulties resting while asleep, poor financial wellbeing, inability to chew certain foods, and fear of solitary death. The GEJE experience was further distinguished in the housing complex group by the loss of an important non-family individual; for the other group, important factors were female gender, junior high school education level or lower, limited social networks, and deterioration of a family member’s health. Older adults’ subjective wellbeing in Soma City was low after nine and a half years following the GEJE. For disaster victims and their families in both groups, it is crucial to implement measures such as long-term, continuous physical and mental health support.

## 1. Introduction

Older adults are more prone to mental health issues such as post-traumatic stress disorder (PTSD) due to natural disasters [1,2,3,4,5], which diminishes their quality of life (QOL) [6,7,8]. The Great East Japan Earthquake (GEJE) struck in March 2011, causing an immense earthquake and tsunami that killed 19,729 people and forced more than 470,000 to evacuate [9]. Furthermore, the Tokyo Electric Power Company’s Fukushima Daiichi Nuclear Power Plant (F1NPP) accident displaced 165,000 residents, 36,000 of whom are still unable to return home (as of March 2021) [9]. Of the total casualties, 66.1% were individuals aged 60 years or above, while 88.6% of disaster-related deaths were those aged 66 years or above [10]. Therefore, countries must plan and implement measures before, during, and after such disasters to reduce their repercussions for older adults [11,12].

Following the complex disaster (GEJE and F1NPP), the Fukushima prefectural government conducted the Fukushima Health Management Survey [13]. According to studies based on survey data, residents of evacuation areas experienced a deterioration in mental health [14], sleep dissatisfaction and excessive drinking [15], and an increase in lifestyle disease risks [16,17]. A different study by Moriyama et al. reported a decline in the social capital of older adults who moved from evacuation areas [18]. Furthermore, Tsubokura reported an increased diabetes risk since 2013 in coastal areas other than evacuation areas, suggesting that the disaster has had secondary effects on residents’ health [19]. Therefore, this study focused on the conditions of older adults in Soma City, which suffered from severe tsunami damage despite not being designated as an evacuation area.

In Soma City, 457 of 458 victims were killed by the tsunami, 5584 houses were damaged, and areas with many casualties were designated as “disaster risk areas” [20]. Based on surveys of disaster victims’ attitudes toward rebuilding their lives and discussions with city residents, Soma City constructed 410 dwellings in nine “public disaster housing complexes” (hereinafter referred to as “housing complexes”) [21,22]. The city undertook this project between 2013 and 2015, earlier than other municipalities, with the intent of sustainably revitalizing the local community and enabling residents to preserve their pre-disaster village communities as well as possible. Priority was given to people who lost their property due to tsunami damage and required government support to rebuild their lives, particularly older adult households. Community connections were fostered among residents, and regular check-ins were also conducted for older adults living alone [21,22]. Although people who did not relocate to the housing complexes also suffered from disaster damage, many remained in the same locations as before and rebuilt their lives on their own. Nine and a half years after the GEJE and 5 years after completing all housing complexes, it became possible to evaluate the QOL of older adults living both inside and outside the housing complexes. Such an evaluation is vital for measuring the effects of reconstruction measures and considering future measures.

The WHO [23] defines QOL as “individuals’ perceptions of their position in life in the context of the culture and value systems in which they live and concerning their goals, expectations, standards and concerns”. There are several proposed QOL frameworks for older adults. For example, Lawton [24] suggests that four components constitute a good life for older adults: behavioral competence, objective environment, psychological wellbeing, and perceived QOL. Perceived QOL is expressed by the degree of satisfaction in various areas measured by subjective wellbeing scales [25], among which Neugarten et al.’s life satisfaction index A (LSIA) [26] and Lawton’s Philadelphia Geriatric Center Morale Scale (PGCMS) [27,28] are widely used. 

This study employed the PGCMS to measure the QOL of older adults. The scale defines morale as a multidimensional concept while calculating it as a one-dimensional score, and has an appropriate length that does not tire the respondent [29]. A high score is considered to indicate “a basic sense of satisfaction for oneself”, “a sense of belonging to one’s environment”, and “an acceptance of unchangeable realities” [30]. Therefore, it was determined to be an effective measure to grasp the situation of disaster victims nine and a half years after the GEJE. Subjective wellbeing is the concept of the superordinate objective variable in this study, which is expressed as “morale” and is based on the PGCMS scores. This is the first study to use the PGCMS to measure the QOL of older adult survivors of disasters including the GEJE. 

Based on findings from his 30-year study, Larson [25] reported that physical health, followed by functional state, economic factors, social interactions, marital status, and lifestyle status, were most strongly associated with subjective wellbeing, while abnormal emergencies were associated with a reduction in wellbeing. Additionally, previous studies on morale reported the following factors related to morale: gender [31,32]; age [33,34]; educational attainment [35]; housing [36]; living with another person or not [35]; higher-level competence, including instrumental activities of daily living [31,34]; health conditions [32,34,35]; sleep conditions [32,33,37]; financial wellbeing [32]; diet variety [37]; chewing ability [36,38]; satisfaction with dietary habits [37]; frequency of communal dining [39,40]; social networks [31,32,34]; perceived loneliness [33,36]; and accumulation of negative life events [40]. 

This study aimed to fulfill two objectives. The first was to compare the subjective wellbeing of older adults living inside and outside housing complexes. Although the GEJE and F1NPP accident most likely reduced the subjective wellbeing of older adults, there were no data on the participants’ PGCMS scores before the GEJE to use for comparison. Therefore, this study compared residents that were collectively relocated to housing complexes with those who rebuilt their lives outside the housing complexes. As Soma City has endeavored to restore and preserve communities in the housing complexes, it was expected that the subjective wellbeing of housing complex residents was close to that of non-housing complex residents. 

The second purpose was to consider factors related to subjective wellbeing and compare them between the two groups. Due to differences in the severity of disaster damage experienced, living environment, and other background elements, the factors related to subjective wellbeing were predicted to differ between the groups. There were no studies that measure subjective wellbeing with PGCMS in post-disaster areas. 

This study addressed the dearth of long-term research on the wellbeing of older adults who live in the complex disaster-stricken areas in Fukushima. Most of the literature in this field to date has been related to general populations that were evacuated from disaster-stricken areas [18,41], while little work has been focused on the older populations that stayed in their hometowns after tsunami damage [42]. This study will contribute to policy, research, and practice regarding public health in disaster-stricken areas. It highlights the benefits of early group relocation [43], serves as a guide to future research on the long-term effects of disasters, and validates the care provided by public health professionals.

Subsequently, this study sought to measure the subjective wellbeing of older adult survivors who collectively relocated to housing complexes, compared to those who did not, and examine the related factors.

## 2. Materials and Methods

### 2.1. Setting

The study site was Soma City, located in the northeastern part of Fukushima Prefecture, approximately 45 km from the Fukushima Daiichi Nuclear Power Plant. The city has a population of 34,631, of which 30.9% were aged 65 years and over as of 29 February 2020 [44]. A tsunami with a height of over 9 m struck the city’s coastal area during the GEJE, flooding 29 km^2^ of land (14.6% of the city’s total area) [45]. 

The survey was conducted in seven town municipalities designated as “land restructuring areas” in the coastal areas (Haragama, Obama, Isobe, Kabaniwa, Niinuma, Hodota, and Babano). There were 114 villages and approximately 8000 residents in the seven towns; however, some areas that were heavily damaged by the tsunami became “disaster risk areas” (off-limit zones), and a total of 410 public disaster housing units and condominiums were constructed in nine new public disaster housing complexes (housing complexes) [21] (Figure 1).

This study employed a cross-sectional design. The survey was conducted between 15 October to 30 November 2020 and took the form of an anonymous, self-administered questionnaire. With the cooperation of the administrative ward mayor (chairperson of the neighborhood association), investigators familiar with the local community visited the residents’ homes, distributed the questionnaire, and collected them 1 to 2 weeks later according to the drop-off and pick-up method.

### 2.2. Participants

The survey targeted individuals aged 65 to 84, and it was estimated that 2503 residents from the seven towns would be potential respondents. The nine villages with housing complexes, which primarily sheltered residents who lost their property due to tsunami damage and required government support to rebuild their lives, were surveyed. Other villages without housing complexes, in which residents generally remained in the same locations as before the GEJE and rebuilt their lives on their own, were randomly selected through cluster sampling; with an effect size of 0.10 [46], a significance level of 0.05, the statistical power of 0.8, and 12 input items, the sample size needed to be at least 184 respondents from both groups to conduct a multivariate regression analysis [47]. Assuming a valid response rate of 60%, the necessary number of participants was estimated to be 309 for the housing complex group (expected collection of 185) and 548 for the non-housing complex group (expected collection of 329). Although the population ratio of housing complex residents to non-housing complex residents was estimated to be 1:7, the survey extracted participants at a ratio of 1:4 to better reflect the population and enable a stratified analysis. For the non-housing complex group, 105 towns without housing complexes were randomly selected through cluster sampling (selection probability of 0.25) until 22 towns with 560 participants were extracted as the survey sample. 

Based on the Basic Resident Register (as of 1 September 2020), a list of survey participants from 31 villages was created. A total of 737 residents from the nine villages with housing complexes, including housing complex residents and general housing residents who lived around the area and 560 residents from the 22 villages without housing complexes, were extracted, amounting to a total of 1297 participants. 

Excluding the 39 participants who were identified as uninvestigable through the administrative ward mayor (eight deceased, 24 who moved or were non-residents, seven in facilities or hospitalized), the questionnaire was distributed to 1258 residents. There were 1133 respondents in total (for a response rate of 87.4%), and after removing 55 incomplete answers, the final number of participants for analysis was 1078 (for a valid response rate of 83.1%). The participants were divided into the “housing complex group” and “non-housing complex group” based on their housing types, not based on their villages. As the disaster experiences of the two groups and their living environments as participants in reconstruction measures were significantly different, they were analyzed separately (Figure 2).

### 2.3. Dependent Variable

The dependent variable was subjective wellbeing, which was measured using the PGCMS total score. The Japanese translated version [48] of Lawton’s revised Philadelphia Geriatric Center Morale Scale [27,28,49] was used for measurement. The scale consists of 17 questions, such as “are you satisfied with your current life?”, to which the respondent answers by selecting either “yes” or “no”. An answer indicating high morale is counted as one point, whereas an answer indicating low morale, or a blank answer, is counted as zero points. The subscales consist of “I: psychological agitation (0–6 points)”, “II: attitude towards aging (0–5 points)”, and “III: loneliness and dissatisfaction (0–6 points)”, with higher points suggesting a higher level of subjective wellbeing. It is generally understood that 13–17 points, 10–12 points, and 0–9 points on the scale signify high, moderate, and low morale, respectively [48].

Based on previous studies, only responses with 12 or more of the 17 questions completed were considered valid [40,48,50,51]. Of the total of 1078 participants for analysis, 957 (88.8%) answered all 17 questions.

### 2.4. Independent Variables

The survey asked questions about 24 variables in six categories related to subjective wellbeing. The demographic variables of gender [31,32], age [33,34], and educational attainment [35] were used as moderator variables (Appendix A). 

#### 2.4.1. Living Environment

The questions asked about the housing type (housing complex or non-housing complex, homeownership) and whether the participants lived with another person. 

#### 2.4.2. Physical Conditions

The degree of independence of higher-level competence was measured using the TMIG Index of Competence [52]. The total score was 13 points, with higher values indicating a greater level of independence. Based on previous studies, the respondents’ results were divided into two groups—0–10 points and 11–13 points [53]. To describe the participants’ basic attributes, the survey also asked about their height and weight (BMI) and whether they had visited a hospital. 

#### 2.4.3. Living Conditions

For health conditions, the survey asked respondents, “how is your current health condition?” They answered using a five-item scale (“very good”, “good”, “normal”, “bad”, and “very bad”) in which the final two options indicated “poor health conditions” For sleep conditions, the survey asked, “over the past month, did you get enough rest while sleeping?” and provided a four-item scale (“sufficient”, “moderate”, “inadequate”, and “none”) in which the final two indicated “difficulties resting while asleep”. For financial wellbeing, the survey asked, “how do you feel about your current financial lifestyle?” The participants responded on a five-item scale (“struggling”, “somewhat struggling”, “normal”, “somewhat comfortable”, and “comfortable”), in which the first two indicate “poor financial wellbeing”.

#### 2.4.4. Dietary Habits

The Dietary Variety Score (DVS) [54] measured food intake diversity. Information was collected about the respondents’ weekly intake of ten food groups. The score ranged from 0 to 10 points, with higher values indicating a larger variety of food intake. Based on previous studies, the respondents’ results were divided into two groups: 0–2 points and 3–10 points [55]. For chewing ability, the survey asked about “situations when respondents chew food” and provided a four-item scale (“can chew any foods”, “cannot chew some foods”, “cannot chew many foods”, and “cannot chew any foods”), in which the final three indicated that the respondent “cannot chew certain foods”. For frequency of communal dining, the survey asked, “how often do you eat with friends, family, relatives, or other individuals?” The participants responded on a six-item scale (“almost every day”, “four-five days per week”, “two-three days per week”, “once per week”, “once or twice per month”, and “rarely”), in which the final option indicated “limited opportunities for communal dining”. For satisfaction with dietary habits, the survey asked, “are you satisfied with your dietary habits (everyday meals)?” The participants responded on a four-item scale (“very satisfied”, “somewhat satisfied”, “not very satisfied”, and “not satisfied”), in which the final two indicated “a lack of satisfaction with dietary habits”.

#### 2.4.5. Community Connections

The degree of social isolation was measured using the Japanese translated version [56] of the simplified Lubben Social Network Scale (LSNS-6) [57,58]. The score results ranged from 0 to 30 points, with higher values indicating a greater social network and any values under 12 indicating social isolation. For fear of solitary death, the survey replicated the Annual Report on the Aging Society [10], asking, “do you consider solitary death (passing away without anyone’s care and being discovered afterward) to be a personally relevant issue?” The participants responded on a five-item scale (“very much”, “somewhat”, “somewhat not”, “not at all”, and “unsure”) in which the first two answers indicated “imminent fear of solitary death”.

#### 2.4.6. Experiences from the GEJE

Regarding the Fukushima Health Management Survey [13], the survey asked respondents to select events they experienced due to the GEJE. The full 12-item list of options was as follows: displacement, living separately from family, living together with family, personal health deterioration, deterioration of a family member’s health, caregiving for a family member, divorce/separation/loss of spouse/partner, loss of a family member other than spouse/partner, loss of an important non-family individual, unemployment, financial hardships, and difficulties in interpersonal relations.

### 2.5. Data Analysis

First, the participants’ characteristics and the distribution of each variable were identified. Next, a univariate analysis was conducted to examine the relationship between the dependent variable, subjective wellbeing (indicated by the PGCMS scores), and all other variables. Continuous variables were analyzed using the *t*-test, and categorical variables were analyzed using Pearson’s chi-squared test.

Of the variables significantly related to subjective wellbeing in these tests, 13 were selected based on Spearman’s rank correlation coefficient between each variable and insight from previous studies. Among the factors that had a high significance in univariate analysis, “deterioration in personal health (GEJE experience)” was similar in content to “poor health condition” (correlation coefficient of ρ = 0.256 for housing complex group, ρ = 0.351 for the non-housing complex group, both *p* < 0.001); therefore, “poor health condition” was selected. “Caregiving for a family member (GEJE experience)” was also similar in content to “deterioration of a family member’s health condition” and was selected. “Caregiving for a family member (GEJE experience)” was also similar in content to “deterioration of a family member’s health (GEJE experience)” (correlation coefficient of ρ = 0.468 for housing complex group, ρ = 0.358 for the non-housing complex group, both *p* < 0.001); thus, “deterioration of a family member’s health” was selected. Furthermore, “Unemployment (GEJE experience)” was similar in content to “financial hardships (GEJE experience)”, which was also similar to “poor financial wellbeing” (housing complex group ρ = 0.491, non-housing complex group ρ = 0.430, both *p* < 0.001); thus, “poor financial wellbeing” was selected. As “a lack of satisfaction with dietary habits” was based on the respondent’s subjective judgment of satisfaction and could be seen as an outcome variable that relates to the same factors as subjective wellbeing, it was not included in the analysis.

A multivariate regression analysis was then conducted, with the independent variables being the binarized data of “no homeownership”, “lack of higher-level competence”, “poor health condition”, “difficulties resting while asleep”, “poor financial wellbeing”, “low DVS”, “cannot chew certain foods”, “limited opportunities for communal dining”, “limited social networks”, “fear of imminent solitary death”, and “experiences from the GEJE (deterioration of a family member’s health, loss of an important non-family individual, difficulties in interpersonal relations)”. The covariate variables were “female”, age (continuous data), and “educational attainment up to junior high school”.

The housing complex and non-housing complex groups were analyzed separately to compare the results. The significance level was set at 0.05. All statistical analyses were conducted using the software IBM SPSS Statistics 27 (IMB Corp., Armonk, NY, USA).

### 2.6. Ethical Considerations

The respondents were informed beforehand of the study purpose and methods, that participation was voluntary, and that the results were entirely anonymous. After a respondent filled out the questionnaire, it was enclosed and sealed in a collection envelope. Questionnaire submission was considered to signal the respondent’s consent to participate. The study was conducted with the approval of the ethics committee of Fukushima Medical University (25 May 2020, approval number: 2020-037).

## 3. Results

### 3.1. Survey Participants’ Characteristics

Table 1 illustrates the survey results for the variables related to the participants’ basic characteristics (Table 1). The total sample was 54.7% women and had a mean age of 73.0 ± 5.4 years old, with no significant difference between the housing complex and non-housing complex groups. The characteristics unique to the housing complex group included a greater proportion of people with an educational attainment of up to junior high school, who lived alone, and who had no homeownership, a low degree of higher-level competence, high BMI, and limited social networks for men. The housing complex group’s mean score was significantly lower (*p* < 0.001), while women’s scores were significantly lower when comparing results between genders (*p* = 0.006, not listed in the table).

### 3.2. Univariate Analysis of PGC Morale

The survey responses for each independent variable were divided according to the two groups, and a Student’s *t*-test was performed on the PGCMS scores (Table 2).

Of the 24 variables, those significantly related to the housing complex group were “higher-level competence”, “health condition”, “sleep conditions”, “financial wellbeing”, “DVS”, “chewing ability”, “satisfaction with dietary habits”, “social networks”, and “fear of solitary death;” from the GEJE experiences, “personal health deterioration”, “deterioration of a family member’s health”, “unemployment”, “financial hardships”, and “difficulties in interpersonal relations”, in addition to the moderator variable “educational attainment”.

The variables significantly related (*p* < 0.05) to the non-housing complex group were “type of housing (homeownership)”, “higher-level competence”, “health condition”, “sleep conditions”, “financial wellbeing”, “DVS”, “chewing ability”, “frequency of communal dining”, “satisfaction with eating habits”, “social networks”, and “fear of solitary death”; from the GEJE experiences, “deterioration of personal health”, “deterioration of a family member’s health”, “caregiving for a family member”, “loss of an important non-family individual”, “unemployment”, “financial hardships”, and “difficulties in interpersonal relations”, in addition to the moderator variables “gender”, “age”, and “educational attainment”.

The variables that had no significant difference between the housing complex and non-housing complex groups were “living alone or with another person”; from the GEJE experiences, “displacement”, “living separately from family”, “living together with family”, “divorce/separation/loss of spouse/partner”, and “loss of a family member other than spouse/partner”.

### 3.3. Multivariate Regression Analysis of PGC Morale

Furthermore, a multivariate regression analysis was conducted on 13 independent variables to examine their relationships with the PGCMS scores (Table 3).

The factors significantly related with the PGCMS scores for both groups were “poor health condition” (housing complex group β = −0.222, non-housing complex group β = −0.263), “difficulties resting while asleep” (housing complex group β = −0.185, non-housing complex group β = −0.229), “poor financial wellbeing” (housing complex group β = −0.341, non-housing complex group β = −0.207), “cannot chew certain foods” (housing complex group β = −0.117, non-housing complex group β = −0.112), and “fear of solitary death”. The related factor was “loss of an important non-family individual (GEJE experience)” (β = −0.125) only for the housing complex group. However, the related factors were “women” (β = −0.099), “educational attainment up to junior high school” (β = −0.062), “limited social networks” (β = −0.113), and “deterioration of a family member’s health (GEJE experience)” (β = −0.076) only for the non-housing complex group. 

The variance inflation factor (VIF) for the housing complex and non-housing complex groups was 1.062–1.061 and 1.043–1.343, respectively. Both values were sufficiently low, and no multicollinearity was observed. The coefficient of determination (R^2^) was 0.391 for both housing complex and non-housing complex groups.

## 4. Discussion

This study examined the subjective wellbeing of older adults in the coastal area of Soma City nine and a half years after the GEJE, in addition to its related factors. Older adults’ subjective wellbeing in Soma City was observed to be at relatively low levels. There were commonalities and differences in the factors related to subjective wellbeing between older adults in the housing complex group and those in the non-housing com-plex group. Both groups had significant levels of satisfaction if the following five needs were addressed: health, sleep, finance, ability to chew, and perceptions of solitary death. Their levels of wellbeing differed depending on individual attributes and relationships with others. These included the loss of friends and acquaintances due to the GEJE for the housing complex group, and gender, educational attainment, social networks, and health of family members for the non-housing complex group.

The mean subjective wellbeing of the survey participants was 9.1 ± 4.4 overall, 8.0 ± 4.6 for the housing complex group, and 9.4 ± 4.3 for the non-housing complex group. Before the GEJE, Nagata et al. [31], who studied Japanese older adults aged 75 years and above, reported morale to be 13.1 ± 2.7 and 12.4 ± 3.0 for men and women, respectively, while Demura et al. [32], who focused on people aged 60 years and above, reported 11.6 ± 3.78 and 11.2 ± 4.02 for men and women, respectively. Lawton explains that scores ranging between 17 to 13, 12 to 10, and 9 to 0 indicate high, medium, and low morale, respectively; therefore, one can infer that the subjective wellbeing of the housing complex group was at relatively low levels [49]. One plausible reason for this decline was the participants’ experiences from the GEJE nine and a half years earlier. In particular, the housing complex group recorded significantly lower subjective wellbeing scores than the non-housing complex group, most likely because they lost their houses and land property to the tsunami. The survey results demonstrate that a greater proportion of the housing complex group experienced “displacement” (housing complex group 67.4%, non-housing complex group 18.2%), “financial hardships” (housing complex group 37.2%, non-housing complex group 24.6%), and “loss of a family member other than spouse/partner” (housing complex group, 21.1%; non-housing complex group, 14.9%) due to the GEJE. Furthermore, Rehdanz et al. [59] reported that the residents who experienced a fall in subjective wellbeing 1 year after the GEJE were those living near tsunami-stricken areas and the F1NPP. This suggests that the experiences of tsunami damage have particularly affected the subjective wellbeing of the housing complex group.

Factors significantly related to subjective wellbeing for both groups were “poor health conditions”, “difficulties resting while asleep”, “poor financial wellbeing”, “cannot chew certain foods”, and “imminent fear of solitary death”. As previous studies have reported similar findings regarding health conditions [32,34,35], sleep conditions [32,33,37], financial wellbeing [2], and chewing ability [37,38], they can be considered as issues shared among older adults in general.

However, the “fear of imminent solitary death” factor should be interpreted with caution due to a lack of previous studies that report its correlation with subjective wellbeing. The term “solitary death” (kodoku-shi in Japanese) first became known when it occurred widely during the 1995 Great Hanshin earthquake [60]. Since then, it has increasingly become a matter of public concern, primarily because of the mass media; however, the term still lacks a solid definition [61]. This study defines it as “passing away without anyone’s care and being discovered afterward” based on the Cabinet Office’s Annual Report on the Aging Society [10]. The government has been implementing measures to ensure that older adults living alone can enjoy life in their community without fearing solitary death. In the 10 years following the GEJE, Fukushima prefecture reported 155 solitary deaths [62], one of which was found in Soma City in 2015 [63]. Despite the local government in Soma City conducting check-in visits for disaster victims, a man in his 50s passed away in temporary housing and was discovered about a week later. The city has since endeavored to prevent similar deaths, particularly in housing complexes, by ensuring that the chairpersons of each administrative ward visit residents about once every 2 days [21]. In the case of the survey participants, although solitary death did not frequently occur in their city, 63.7% of the housing complex group and 52.6% of the non-housing complex group subjectively felt that solitary death was an issue of personal relevance. The survey results also indicated that fear of solitary death was associated with lower levels of subjective wellbeing. Future prevention measures should focus on supporting men and residents living alone, who are most prone to solitary death. Implementing better methods to monitor people’s health, such as a robust family doctor system and the active use of caregivers, will also be necessary [64].

Older adults are prone to conditions such as tooth loss, tooth decay, periodontal disease, and xerostomia, resulting in reduced chewing ability [65]. Chewing ability is closely related to older adults’ QOL [38]. Ohara reported the effectiveness of educational programs such as oral hygiene instruction, facial and tongue muscle exercises, and salivary gland massage for xerostomia [66]. Katagiri reported the efficacy of mastication training using ice chips [67]. Therefore, guidance by dentists and dental hygienists is important [65]. In addition, since elderly people with oral function problems tend to have low multiple nutrient intake [68], nutritional guidance by registered dietitians, including assessments of dietary status and suggestions on how to eat, is also considered necessary.

The factor “loss of an important non-family individual (GEJE experience)” only had a strong correlation with subjective wellbeing for the housing complex group. This may be because many housing complex residents, who had lived in coastal settlements in which neighbors closely supported each other, lost valuable people and community connections due to the tsunami. Kun et al. [4] reported that people in areas with greater earthquake damage were at a higher risk of post-traumatic stress disorder (PTSD). Additionally, Jia et al. [5] reported that risk factors for PTSD symptoms in earthquake survivors include older age, loss of family members, a sense of guilt over someone’s death or injury, and lack of mental health support. Thus, it is crucial that residents who experienced the GEJE, many of whom live in housing complexes, receive continuous professional mental health support. 

On the contrary, the factors “female”, “educational attainment up to junior high school”, “limited social networks”, and “deterioration of a family member’s health (GEJE experience)” only had a strong correlation with subjective wellbeing for the non-housing complex group. Nagata et al. [31] and Demura et al. [32] both indicate that the female gender is related to low levels of morale; however, Hamashima [69] suggests that gender differences are not evident for older age groups. The “female” in the non-housing complex group may have low levels of subjective wellbeing because the residents live in conditions similar to those of the average Japanese resident. The low morale level associated with educational attainment was previously shown by Hamashima [69] and Iwasa et al. [35]. Regarding social networks, low subjective wellbeing was associated with social isolation due to fewer opportunities to interact with family members, relatives, and friends. Nagata et al. [31], Okamoto [34], and Demura et al. [32] concur, reporting “frequency of socializing opportunities”, “family conversations”, and “number of close friends” as factors related to subjective wellbeing. Although there was no observable relationship between social networks and subjective wellbeing for the housing complex group, the quantity and quality of social networks of older adults are greatly significant [33]. Subjective wellbeing can be improved by building various social connections with people, including friends and family members [70], which suggests that community development must be supported in non-housing complex areas as well. As the GEJE experience factor “deterioration of a family member’s health” had no relation with the factors “divorce/separation/loss of spouse/partner” and “loss of a family member other than spouse/partner”, the struggles of caring for unwell family members may affect the subjective wellbeing of older adult GEJE survivors, separate from the direct grief of losing a family member. Therefore, it is vital to provide long-term, continuous physical and mental health support to disaster victims and their families. 

The results of this study suggest that health conditions, sleep conditions, financial wellbeing, chewing ability, and fear of solitary death may be related to older adults’ subjective wellbeing in disaster-stricken areas. Additionally, the loss of an important non-family individual may be related to subjective wellbeing for housing complex residents (consisting mostly of individuals who experienced severe tsunami damage), while gender, educational attainment, social networks, and the deterioration of a family member’s health may relate to subjective wellbeing for residents outside housing complexes. The difference in related factors between the housing complex and non-housing complex groups indicates that Soma City’s housing complexes have helped maintain social networks by revitalizing community connections. In future instances of group relocation in disaster-stricken areas, housing complexes should be developed to preserve pre-existing village communities as much as possible. 

This study had several limitations. First, it followed a cross-sectional design and thus did not illustrate any causal relationships. Second, as the study did not measure PGCMS scores before the GEJE, it cannot be denied that participants’ subjective wellbeing levels may have been low before the GEJE. In the future, surveys of PGCMS scores in other disaster-stricken areas should be conducted. Despite the above limitations, the study used data that had a high response rate (87.4%) and were representative of the region. For the first time, nine and a half years after the GEJE, the study successfully revealed the subjective wellbeing of older adults affected by the earthquake, tsunami, and F1NPP accident in Soma City.

Finally, there may be limitations to the generalizability of this study’s findings, because it was conducted in a single region in Japan. However, the high response rate suggests that the findings reflected the reality in that region. Similar disasters will likely occur in other parts of the world in the future (e.g., tsunamis, complex disasters, etc.), and the findings may be generalizable to regions with similar geographic conditions and social backgrounds. The findings, therefore, may help those areas develop supportive measures.

## 5. Conclusions

The subjective wellbeing of older adults in Soma City was observed to be at relatively low levels nine and a half years after the community suffered severe damage from the earthquake, tsunami, and F1NPP accident in March 2011.There were similarities and differences in the related factors for the housing complex and non-housing complex groups. These results indicate that, for both groups, it is crucial to implement measures such as long-term, continuous physical and mental health support for disaster victims and their families; better welfare support and family doctor systems, as well as encouragements to use caregivers actively; communication within local communities; and oral health guidance and nutrition support. The results also suggest that continuous professional support for mental health and greater community development are vital for the housing complex and non-housing complex groups, respectively.

## Figures and Tables

**Figure 1 ijerph-19-02639-f001:**
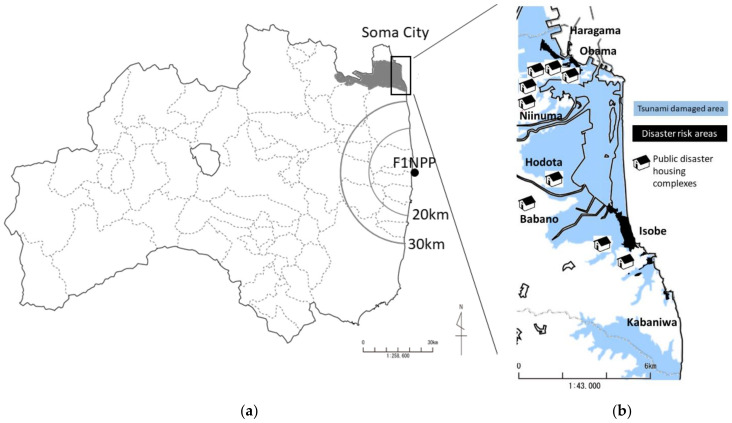
(**a**)The geographical location of Soma City in Fukushima Prefecture, (**b**) illustration of its coastal area (areas that suffered tsunami damage, disaster risk areas, and public housing complexes).

**Figure 2 ijerph-19-02639-f002:**
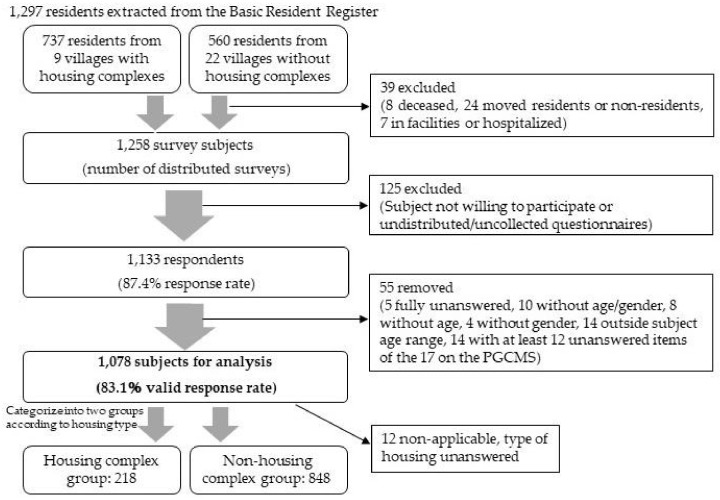
A flow chart illustrating the survey procedure.

**Table 1 ijerph-19-02639-t001:** Participant characteristics.

		Housing Complex Group	Non-Housing Complex Group	Overall	*p*-Value
		*n* = 218	*n* = 848	*n* = 1066	
Sex	Women (%)	119 (54.6)	464 (54.7)	583 (54.7)	0.973
Age	Mean age (SD)	73.0 (5.3)	73.0(5.4)	73.0 (5.3)	0.931
Educational level	Up to junior high school (%)	137 (64.3)	350 (42.4)	487 (46.9)	<0.001
Household composition	Living alone (%)	85 (39.7)	145 (17.3)	230 (21.8)	<0.001
Type of housing	No homeownership (%)	153 (70.2)	117 (13.8)	796 (74.7)	<0.001
Physical condition	Mean higher-level competence (0–13 points) (SD)	10.2 (2.7)	10.7 (2.6)	10.6 (2.7)	0.022
	Mean BMI kg/m^2^ (SD)	24.4 (3.5)	23.6 (3.5)	23.8 (3.5)	0.007
	Regularly visits a hospital (%)	170 (80.2)	681 (82.6)	851 (82.1)	0.405
Social networks mean (SD)	Total (0–30 points) Overall	12.1 (6.1)	12.8 (6.0)	12.6 (6.0)	0.169
	Men	9.9 (6.4)	12.5 (6.2)	11.9 (6.3)	<0.001
	Women	14.0 (5.2)	13.0 (5.8)	13.2 (5.7)	0.107
Subscales	I Family and relatives (0–15 points)	6.6 (3.4)	7.2 (3.2)	7.1 (3.2)	0.012
	Ⅱ Friends and acquaintances (0–15 points)	5.6 (3.6)	5.6 (3.6)	5.6 (3.6)	0.968
Subjective wellbeing (SD)	Total (0–30 points) Overall	8.0 (4.6)	9.4 (4.3)	9.1 (4.4)	<0.001
	Men	8.2 (4.7)	9.9 (4.2)	9.5 (4.4)	0.001
	Women	7.9 (4.5)	9.0 (4.3)	8.8 (4.4)	0.011
Subscales	Ⅰ Psychological agitation (0–6 points)	3.3 (2.1)	3.7 (1.9)	3.6 (1.9)	0.017
	Ⅱ Attitude towards aging (0–5 points)	2.1 (1.6)	2.5 (1.6)	2.4 (1.6)	0.001
	Ⅲ Loneliness and dissatisfaction (0–6 points)	2.6 (1.7)	3.2 (1.6)	3.1 (1.7)	<0.001
GEJE experiences (%)	Displacement	147 (67.4)	154 (18.2)	301 (28.2)	<0.001
	Living separately from family	47 (21.6)	75 (8.8)	122 (11.4)	<0.001
	Living together with family	17 (7.8)	76 (9.0)	93 (8.7)	0.587
	Deterioration of personal health	71 (32.6)	190 (22.4)	261 (24.5)	0.002
	Deterioration of a family member’s health	36 (16.5)	99 (11.7)	135 (12.7)	0.055
	Caregiving for a family member	30 (13.8)	88 (10.4)	118 (11.1)	0.155
	Divorce/separation/loss of spouse/partner	25 (11.5)	65 (7.7)	90 (8.4)	0.072
	Loss of a family member other than spouse/partner	46 (21.1)	126 (14.9)	172 (16.1)	0.025
	Loss of an important non-family individual	86 (39.4)	275 (32.4)	361 (33.9)	0.051
	Unemployment	37 (17.0)	52 (6.1)	89 (8.3)	<0.001
	Financial hardships	81 (37.2)	209 (24.6)	290 (27.2)	<0.001
	Difficulties in interpersonal relations	25 (11.5)	57 (6.7)	82 (7.7)	0.019

**Table 2 ijerph-19-02639-t002:** Results of univariate analysis of independent variables and subjective wellbeing (PGCMS scores).

		Housing Complex Group				Non-Housing Complex Group			
		Frequency	Mean	SD	*p*-Value	Frequency	Mean	SD	*p*-Value
Gender	Men	99	8.2	4.7	0.558	384	9.9	4.3	0.005
	Women	119	7.9	4.5		464	9.2	4.4	
Age	65–74	141	7.9	4.8	0.568	549	9.7	4.4	<0.001
	75–84	77	8.3	4.3		299	8.9	4.2	
Educational level	Above high school graduation	76	9.0	4.8	0.021	475	9.9	4.2	<0.001
	Up to junior high school	137	7.5	4.5		350	8.8	4.3	
Type of housing	Homeownership	65	8.4	4.7	0.404	731	9.6	4.3	0.001
	No homeownership	153	7.9	4.5		117	8.2	4.1	
Household composition	Living together	129	8.0	4.9	0.718	694	9.4	4.3	0.470
	Living alone	85	8.2	4.2		145	9.2	4.3	
Physical condition Higher-level competence									
	0–10 points: Low group	93	7.2	4.2	0.027	288	8.1	4.4	<0.001
	11–13 points: High group	117	8.7	4.8		541	10.1	4.1	
Health condition	Very good, good, normal	161	9.1	4.4	<0.001	672	10.5	3.9	<0.001
	Bad, very bad	57	5.0	3.8		167	5.4	3.5	
Resting during sleep	Sufficient, moderate	177	8.7	4.5	<0.001	713	10.1	4.1	<0.001
	Inadequate, none	37	4.8	4.0		121	5.4	3.6	
Financial wellbeing	Comfortable—normal	108	9.9	4.4	<0.001	546	10.5	4.0	<0.001
	Somewhat struggling, struggling	104	6.1	4.0		278	7.3	4.0	
Dietary habits DVS	0–2 points: Low group	105	7.2	4.6	0.010	366	8.8	4.3	<0.001
	3–10 points: High group	110	8.8	4.5		476	9.8	4.2	
Chewing ability	Can chew any foods	108	9.0	4.7	0.003	473	10.4	4.1	<0.001
	Cannot chew some—any foods	105	7.1	4.4		353	8.2	4.3	
Frequency of communal dining									
	At least once per month	163	8.3	4.7	0.223	715	9.7	4.2	<0.001
	Almost never	53	7.4	4.1		129	7.6	4.4	
Satisfaction with dietary habits									
	Very satisfied, somewhat satisfied	154	9.2	4.5	<0.001	707	10.1	4.1	<0.001
	Not very satisfied, not satisfied	59	5.0	3.3		137	6.0	3.9	
Social networks	0–11 points: Low group	93	7.3	4.6	0.034	339	8.1	4.4	<0.001
	12–30 points: High group	124	8.6	4.5		507	10.2	4.1	
Fear of solitary death	Not at all, somewhat not, unsure	78	9.0	4.8	0.021	397	10.0	4.1	<0.001
	Very much, somewhat	137	7.5	4.4		440	8.8	4.4	
Experiences from the GEJE ^1^									
Displacement		147	8.0	4.7	0.713	154	9.0	4.2	0.268
Living separately from family		47	7.3	4.1	0.191	75	8.8	3.9	0.217
Living together with family		17	9.2	4.1	0.261	76	9.6	4.1	0.666
Deterioration of personal health		71	5.7	3.8	<0.001	190	7.0	4.1	<0.001
Deterioration of a family member’s health		36	6.0	4.2	0.003	99	7.6	4.4	<0.001
Caregiving for a family member		30	7.2	4.7	0.267	88	8.0	4.8	0.004
Divorce/separation/loss of spouse/partner		25	8.2	5.1	0.883	65	8.5	4.1	0.087
Loss of a family member other than spouse/partner		46	7.1	4.7	0.125	126	9.0	4.2	0.220
Loss of an important non-family individual		86	7.7	5.0	0.355	275	9.0	4.2	0.047
Unemployment		37	6.6	4.6	0.036	52	8.1	4.2	0.031
Financial hardships		81	6.2	4.0	<0.001	209	7.7	4.2	<0.001
Difficulties in interpersonal relations		25	4.6	3.9	<0.001	57	7.9	4.4	0.007

^1^ Multiple answers; the test assesses respondent’s experience.

**Table 3 ijerph-19-02639-t003:** Factors related to Subjective wellbeing.

	Housing Complex Group *n* = 205			Non-Housing Complex Group *n* = 781		
	Beta ^1^	*p*-Value	95% Confidence Intervals	Beta ^1^	*p*-Value	95% Confidence Intervals
Age (continuous value)	−0.057	0.341	−0.174–0.060	−0.056	0.058	−0.114–0.002
Gender: Female	−0.087	0.153	−0.207–0.033	−0.099	0.001	−0.156–−0.043
Educational level: Up to junior high school	−0.046	0.472	−0.170–0.079	−0.062	0.033	−0.120–−0.005
Living environment: No homeownership	−0.046	0.424	−0.160–0.068	−0.013	0.662	−0.071–0.045
Physical conditions						
Lack of higher-level competence	0.046	0.509	−0.091–0.183	−0.044	0.177	−0.108–0.020
Living conditions:						
Poor health condition	−0.222	0.001	−0.350–−0.094	−0.263	<0.001	−0.324–−0.203
Difficulties resting while asleep	−0.185	0.003	−0.305–−0.065	−0.229	<0.001	−0.288–−0.170
Poor financial wellbeing	−0.341	<0.001	−0.461–−0.221	−0.207	<0.001	−0.266–−0.148
Dietary habits:						
Low DVS	−0.088	0.177	−0.215–0.040	−0.033	0.270	−0.090–0.025
Cannot chew certain foods	−0.117	0.047	−0.233–−0.002	−0.112	0.000	−0.169–−0.054
Limited opportunities for communal dining	−0.069	0.284	−0.195–0.057	−0.025	0.398	−0.084–0.034
Community connections:						
Limited social networks	−0.001	0.983	−0.132–0.129	−0.113	<0.001	−0.175–−0.051
Fear of imminent solitary death	−0.171	0.003	−0.283–−0.059	−0.075	0.009	−0.131–−0.019
GEJE experiences:						
Deterioration of a family member’s health	−0.117	0.051	−0.234–0.000	−0.076	0.010	−0.133–−0.018
Loss of an important non-family individual	−0.125	0.040	−0.245–−0.006	−0.051	0.080	−0.107–−0.018
Difficulties in interpersonal relations	−0.090	0.131	−0.207–0.027	0.016	0.575	−0.040–0.006
Adjusted coefficient of determination	0.391			0.391		

^1^ Standardized regression coefficients.

## Data Availability

Not applicable.

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
