# Peer review of "Subjective Wellbeing and Related Factors of Older Adults Nine and a Half Years after the Great East Japan Earthquake: A Cross-Sectional Study in the Coastal Area of Soma City"

_ijerph, 2022, doi:10.3390/ijerph19052639_

Round 1

Reviewer 1 Report

The reviewed article takes up a very important and up-to-date problem. Authors applied the well-known PGCMS total score  to assess the quality of life of the elderly. The novelity of the paper lies in applying already known methodology to a unique research sample - individuals aged 65 to 84 who suffered Great East Japan Earthquake.

The analysis is based on dataset with 1078 respondents. The advantage of the article is a very detailed description of the method of selecting the research sample, which is not a common practice in this type of research.

The methodology based on univariate analysis and multivariate regression is appropriate.

Below are some suggestions which I recommend the authors to consider

  • In the introduction, I suggest to emphasize (even more) the contribution of the article and its uniqueness in comparison with the literature to date.
  • A "study design" paragraph with just one sentence looks unusual. I would consider removing it as it adds nothing new to the paper.
  • There are some missing information in lines 289-290.
  • In the “Results” section the presentation of the results is very direct and mechanically duplicates the information contained in the tables. Generally, this is not a mistake, but it may make the reader lose interest in further reading. Adding some in-depth analysis at this point would improve the quality of the text.
  • Although the authors discuss individual questions from the questionnaire in sections 2.5 and 2.6, this approach may be still confusing for readers. I suggest to additionally (e.g. in an appendix or in supplementary materials) include the exact wording of the questions asked to the respondents.

Author Response

Dear reviewer #1

We wish to express our appreciation to the Reviewer #1 for his or her insightful comments, which have helped us significantly improve the paper.

Kind regards,

Yuri Kinoshita

Reviewer 2 Report

Thank you for this review invitation. This article is well written and  an important article in this field. However few points suggested to the authors for further respond:

The authors stated that the limitation of this study is in its unclarity of the related factors affect subjective wellbeing or, conversely, low subjective wellbeing levels cause various effects, however the authors concluded it in the contrary statement by saying that it observed to be relatively low levels of subjective wellbeing. Please clarify and rephrase to avoid potential misleading conclusion by the readers.

In line 426, the authors mentioned that 'Chewing ability' aspect as part of major findings, I would suggest to the authors to elaborate this exhaustively with larger references, as I believe oral health (oro-motor) is part of the key health indicator in ageing population. 

On the 'Result section', I would suggest to the authors to simplify (shortened) the presentation by only showing the most relevant and statistically significant variables, especially in multivariate result. Let the table talk by itself.

It would be interesting to know if this model of care for a traumatised community could be  applied in other cultural settings and with a wider age demographic.

Author Response

Dear reviewer #2

We wish to express our appreciation to the Reviewer #2 for his or her insightful comments, which have helped us significantly improve the paper.

Kind regards,

Yuri Kinoshita
